# Combined analysis of transposable elements and structural variation in maize genomes reveals genome contraction outpaces expansion

Manisha Munasinghe[1]*, Andrew Read[1,2], Michelle C. Stitzer[3,4], Baoxing Song[5], Claire C. Menard[2], Kristy Yubo Ma[6], Yaniv Brandvain[1,7], Candice N. Hirsch[2], Nathan Springer[1]

1 Department of Plant and Microbial Biology, University of Minnesota, St. Paul, Minnesota, United States of America, 2 Department of Agronomy and Plant Genetics, University of Minnesota, St. Paul, Minnesota, United States of America, 3 Institute for Genomic Diversity, Cornell University, Ithaca, New York, United States of America, 4 Department of Molecular Biology and Genetics, Cornell University, Ithaca, New York, United States of America, 5 Peking University Institute of Advanced Agricultural Sciences, Weifang, China, 6 Department of Mathematics, Statistics, and Computer Science, Macalester College, St. Paul, Minnesota, United States of America, 7 Department of Ecology, Evolution and Behavior, University of Minnesota, St. Paul, Minnesota, United States of America

* mmunasin@umn.edu

**Data Availability Statement:** All datasets used in this analysis are publicly available. TE annotations, gene annotations, and gene synteny calls were

## Abstract

Structural differences between genomes are a major source of genetic variation that contributes to phenotypic differences. Transposable elements, mobile genetic sequences capable of increasing their copy number and propagating themselves within genomes, can generate structural variation. However, their repetitive nature makes it difficult to characterize fine-scale differences in their presence at specific positions, limiting our understanding of their impact on genome variation. Domesticated maize is a particularly good system for exploring the impact of transposable element proliferation as over 70% of the genome is annotated as transposable elements. High-quality transposable element annotations were recently generated for *de novo* genome assemblies of 26 diverse inbred maize lines. We generated base-pair resolved pairwise alignments between the B73 maize reference genome and the remaining 25 inbred maize line assemblies. From this data, we classified transposable elements as either shared or polymorphic in a given pairwise comparison. Our analysis uncovered substantial structural variation between lines, representing both simple and complex connections between TEs and structural variants. Putative insertions in SNP depleted regions, which represent recently diverged identity by state blocks, suggest some TE families may still be active. However, our analysis reveals that within these recently diverged genomic regions, deletions of transposable elements likely account for more structural variation events and base pairs than insertions. These deletions are often large structural variants containing multiple transposable elements. Combined, our results highlight how transposable elements contribute to structural variation and demonstrate that deletion events are a major contributor to genomic differences.

downloaded from MaizeGDB. TE and gene annotations were downloaded from https://maizegdb.org/NAM_project, while synteny classifications for the NAM genes were download from https://ars-usda.app.box.com/v/maizegdb-public/folder/186350887665. All remaining datasets have been deposited on Dryad and can be accessed at https://doi.org/doi:10.5061/dryad.5qfttdz9t. This includes the pairwise AnchorWave alignments in both MAF and GVCF format, summarized AnchorWave outputs, TE and gene classification calls, and the sliding window SNP counts used to identify SNP depleted regions. Scripts used to filter and analyze data are available on GitHub at https://github.com/mam737/PolymorphicTEs_NAM. To visualize pairwise alignments with overlapping TE and gene annotations, an R Shiny App Web Browser was developed and can be found at https://mmunasin.shinyapps.io/nam_sv/.

**Funding:** This material is based upon work supported by the NSF Postdoctoral Research Fellowship in Biology under Grant No. IOS-2010908 (M.M.), Grant No. IOS-2109697 (A.R.), and Grant No. IOS-1907343 (M.C.S.) which provided salary for each of the awardees. It was also supported in part by NSF Grant No. IOS-1934384 (N.S.). The funders had no role in study design, data collection and analysis, decision to publish, or preparation of the manuscript.

**Competing interests:** The authors have declared that no competing interests exist.

## Author summary

Mutation followed by selection, in addition to random genetic drift, is the basis for evolution and drives diversification. These mutations can be small single nucleotide changes or larger insertions or deletions of DNA sequence, referred to as structural variants. Structural variation is often associated with the repetitive nature and functional activity of transposable elements. Here, we used whole genome alignments to identify structural variation between the maize reference genome and 25 other inbred maize lines. We intersected these genomic structural variants with publicly available transposable element annotations to determine what proportion of the structural variation is annotated as transposable element. The composition of transposable elements within each structural variant allowed us to infer whether the variant represented a likely insertion or deletion event, especially within recently diverged SNP depleted regions. We find that many variants are likely deletion events composed of multiple transposable element types, as well as sequence that did not originate from a transposable element. Our results highlight the complex relationship between transposable elements and genomic diversity, and we provide a valuable resource for future studies of diversity across transposon-rich maize genomes.

## Introduction

Plant genomes are replete with transposable elements (TEs)—accounting for as little as 20% of the genomes of *Medicago truncatula* and *Arabidopsis thaliana* [1] to over 70% of the genomes of maize and wheat [1–3]. TE expansion is mediated by the active movement of TEs, particularly class I 'copy-and-paste' elements that utilize an RNA intermediate, and can contribute to expansions in genome size. Many closely related plant species have similar gene content but substantial differences in genome size attributable to TE accumulation in one species [4–7]. The continued movement of TEs in many plant lineages has been hypothesized to lead to a 'one-way ticket to genome obesity' [8]. Ongoing deletion events in many plant genomes can counteract genome size expansion caused by TE accumulation [8–13]. Several studies have explored the influence of both TE expansion and ongoing deletion events on genome size variation [14–16], and they have been instrumental in understanding how TEs shape genome evolution. However, these studies were limited in scope due to technological limitations.

Advancements in whole-genome long-read sequencing and computation methods have rapidly enhanced our ability to characterize and investigate structural variation between genomes [17–19]. Genome-wide characterization of structural variation in the maize genome has found extensive variability in genome sequence [20–26], gene content [25–28], and transposable element content [26,29]. Detailed characterization of multiple haplotypes for several loci in maize revealed extensive structural polymorphisms for TE content [30–32]. Given the high TE content of the maize genome [2], it is likely that transposable elements are a major contributor to structural variants (SVs), but this has yet to be fully quantified.

The recently completed high-quality genome sequences of the 26 maize inbred lines used to generate the nested association mapping (NAM) population provides an opportunity to generate a high-resolution understanding of transposable element polymorphisms and the extent to which variation in TEs contributes to SVs and phenotypic variation in maize [26]. These lines were selected from a larger association panel to provide a sampling of maize diversity. As such, these lines have limited genetic relatedness or population structure. We generated base-pair

resolved whole genome alignments in a pairwise fashion by aligning the B73 reference genome to each of the other NAM founder lines using AnchorWave [33]. We developed an approach that intersected these pairwise alignments with robust, consistent TE annotations generated for each of the NAM lines [26,34]. This allowed us to classify each TE annotation as either shared, polymorphic, or ambiguous between two genomes. Applying this approach revealed that >30% of TEs are polymorphic in comparisons between B73 and a given NAM genome. A comparison of all structural variants and TE annotations revealed that TEs contribute substantially to structural variation among NAM genomes, but that only a subset of the structural variants have features that suggest simple TE insertion polymorphisms. A careful examination of large genomic regions that are likely recently diverged across these comparisons identified a subset of TE families that may have ongoing movement in modern maize inbreds and allowed estimates of ongoing insertion / deletion rates.

## Results

Our first goal was to quantify the number of TEs in maize genomes and how these TEs contribute to genome size. To do so, we made use of previous annotations generated using the panEDTA approach [26,34,35]. This provided a set of structural annotations representing putative full-length transposons with intact structural features (long terminal repeats, terminal inverted repeats, target site duplications, etc.) as well as homology based annotations of transposon-associated sequences that contain sequence similarity to structurally annotated elements. After numerous standard quality control steps (e.g., excluding helitron annotations–which are quite difficult to accurately annotate [29,34,36], non-TE repeats, and potentially misannotated TEs), we found that, on average, NAM lines had 858,902 transposable elements. 7.9% of these TEs were structurally annotated and the remaining 92.1% relied on homology-based annotations (S1 Table). The average total of transposable element sequence in the NAM genomes was 1,655Mb (min = 1,634Mb, max = 1,673Mb) which accounts for ~77% of the total genome size. The structurally annotated TEs accounted for an average of 31.2% of the total TE Mb (S1 Table). In contrast, only 61Mb of the NAM genomes (2.9% of the genome) was annotated as genes. Structural annotations accounted for 8.7% of all class I elements and 6.3% of all class II elements. However, 32.8% of the Mb of class I elements were structurally annotated, while only 14.1% of the Mb of class II elements were structurally annotated.

There are many difficulties in accurately annotating TEs that can complicate the exact quantification of TE variation. For example, homology annotations or nested insertions can result in a single TE being represented by multiple annotation fragments that are not clearly associated with a single element. This can lead to the number of TE annotations over-representing the total number of actual TEs. This is particularly noted for longer LTR elements. While the true number of shared and polymorphic transposons can vary based on annotation quality and approaches, the cumulative base pairs of TEs that are shared or polymorphic are less subject to influences based on annotating fragments of a TE. Consequently, we report both the number and cumulative Mb of TE-associated sequences.

### Highly variable TE content among NAM genomes

We next aimed to characterize variation in TE content across maize lines. Pairwise whole-genome alignments generated using AnchorWave [33] were used to classify TEs as shared or polymorphic between B73 and a singular NAM genome. In each pairwise contrast between B73 and a NAM genome, each region of the alignment could be classified into one of three types: alignable sequence, structural variant sequence present in one genotype relative to the other genotype, or unalignable sequence (see methods for details) (Fig 1A). Across all pairwise

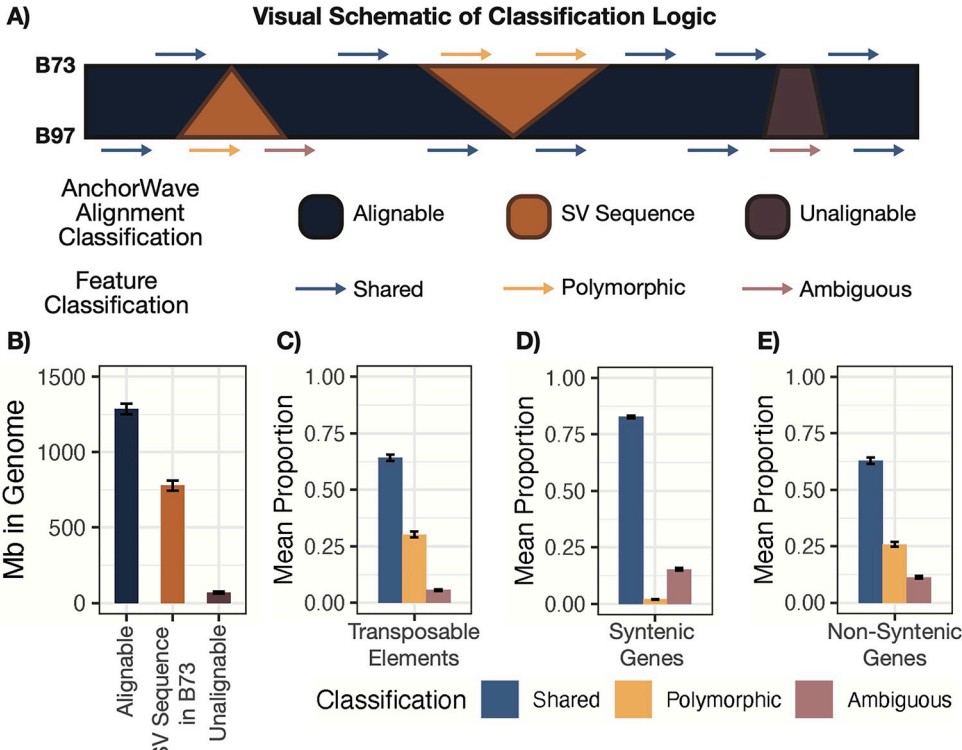

**Fig 1. Proportions of Various Classifications Across B73 versus NAM comparisons.** (A) A visual schematic describing the logic flow used to both partition the pairwise AnchorWave alignments between B73 and every other NAM line into alignable, structural variant, or unalignable sequence and to intersect these regions with feature annotations to classify features as either shared, polymorphic, or ambiguous. (B) Barplots showing the Mb of sequence in B73 classified as alignable (dark blue), structural variant sequence in B73 (orange), and unalignable (purple) across all pairwise comparison between B73 and every other NAM genome. Barplots showing the mean proportion of (C) transposable elements, (D) syntenic genes, and (E) non-syntenic genes in B73 classified as either shared (light blue), polymorphic (yellow), or ambiguous (light purple) across all pairwise comparisons between B73 and every other NAM genome. Height of the bar in panels B–E indicate the mean with error bars indicating the standard deviation across all 25 pairwise comparisons.

comparisons between B73 and all other NAM lines, 60.3% of the B73 genome was alignable (min = 57.6%, max = 63.6%), 36.5% was inserted sequence in B73 (min = 33.1%, max = 38.8%), and 3.2% was unalignable (min = 2.7%, max = 3.8%) when compared to the other genomes (Figs 1B and S1). Reassuringly, on average, 36.1% of a NAM's genome contained inserted sequences absent in B73 (S1 Fig).

To classify TEs, the coordinates of annotated TEs within NAM genomes were compared to the AnchorWave classifications of the genome. Any TE that had ≥ 95% overlap with alignable regions was classified as 'shared' between B73 and the focal NAM line, while TEs that had ≥ 95% overlap with structural variant sequence were classified as 'polymorphic' between B73 and the focal NAM line. The remaining TE annotations were classified as 'ambiguous'. These ambiguous features included examples that were within unalignable regions as well as those that partially overlapped (< 95% overlap) regions with alignable or inserted sequence (*i. e.*, ambiguous features are those that do not meet the threshold to be either shared or polymorphic). On average, 64.2% of B73 TEs were classified as shared, 30.2% were classified as polymorphic, and the remaining 5.6% were classified as ambiguous (Fig 1C). This means that, in any pairwise comparison between B73 and another NAM genome, approximately 259,000 B73 TE annotations were polymorphic or absent in the NAM genome. There was relatively little

variation in the number of TEs that were classified as either shared, polymorphic, or ambiguous depending on whether we were characterizing the TEs present in B73 or the compared NAM genome, and the proportions were quite similar for each of the NAM genomes (S2 Fig).

Several prior studies have evaluated the frequency of present-absent gene sequences among maize genomes using a variety of approaches [21,23,25–27]. To evaluate the frequency of shared and polymorphic classifications for TEs relative to genes in a consistent fashion, we applied the same approach described for TEs above to gene annotations in maize (Fig 1D and 1E). The maize gene annotations were split into syntenic and non-syntenic (based on comparisons to other grasses). Non-syntenic genes often include pseudogenes or transposed gene fragments, and they are much more variable between genomes [37]. As expected, we found that syntenic genes were much more likely to be shared between genomes with a relatively low rate of polymorphic cases (Fig 1D). Non-syntenic genes, in contrast, have higher rates of polymorphic genes that were nearly as high as the rate of polymorphic TEs (Fig 1E). The frequencies of polymorphic genes based on our approach was similar to previous estimates [26]. In general, genes exhibited a higher frequency of ambiguous classifications in comparison to TEs, but many of these likely reflect insertion/deletion events within introns that result in less than 95% of the gene sequence being present within alignable sequence. When we classified genes based only on exon sequence, we found that the proportion of ambiguous classifications was reduced (S2 Fig).

The B73 TEs were compared to each of the other NAM genomes in pairwise comparisons. An analysis across each of these pairwise comparisons provided the opportunity to characterize the classification frequency for each of the B73 TEs (Fig 2A). The B73 features, either TEs

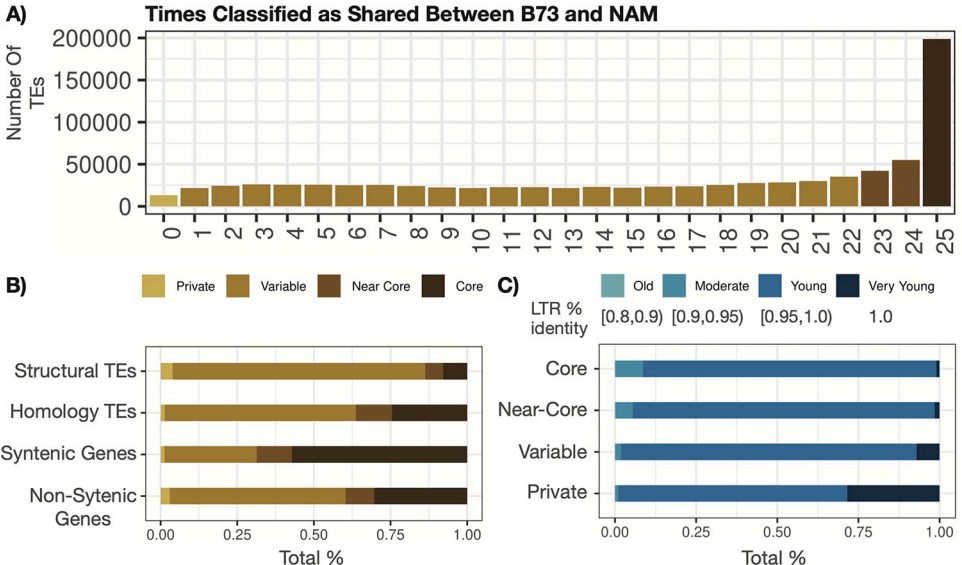

**Fig 2. Consistency of B73 Feature Classifications Across B73 versus NAM comparisons.** Every feature in the genome (*i.e.*, TE or gene annotation) is classified as shared, polymorphic, or ambiguous and binned into either core (classified as shared across all 25 comparisons–dark brown), near-core (classified as shared in 23–24 comparisons–brown), variable (classified as shared in 1–24 comparisons–light brown), or private (never classified as shared across all comparisons–tan). (A) Distribution of all B73 TEs across all NAM comparisons. (B) The proportion of B73 structurally-annotated TEs, homology-annotated TEs, syntenic genes, and non-syntenic genes classified as either core, near-core, variable, or private. (C) The age of a structurally-annotated LTR was estimated from the percent identity between the two long terminal repeats and binned into either very young (% identity = 1), young (% identity within [0.95,1)), moderate (% identity within [0.9,0.95)), and old [0.8,0.9)) with darker colors representing younger TEs. B73 structural TEs were partitioned depending on whether they are core, near-core, variable, or private and the proportion of each category classified as young, very young, moderate, or old is shown.

or genes, could be classified as core (classified as shared across all 25 comparisons), near-core (classified as shared in 23–24 comparisons), variable (shared in 1–22 comparisons), or as private (never classified as shared across all 25 comparisons) (Fig 2A). A comparison of the TEs that were annotated as structurally intact or homology-based revealed that structurally annotated TEs were depleted for core and near-core features, exhibiting a higher frequency of variable and private elements (Fig 2B). Non-syntenic genes exhibited a distribution of classifications that were quite similar to the homology-annotated TEs, while the syntenic genes were enriched for core features with relatively few variable and private annotated features (Fig 2B). This was consistent with the observation that, in maize, many non-syntenic genes exist within or substantially overlap TEs [29].

Structurally annotated LTR elements could be used to assess the relative age of TEs that were classified as either core or variable. The two long terminal repeats are 100% identical at the time of insertion due to the mechanism of LTR TE mobilization. Older elements will accumulate polymorphisms that can distinguish the two long terminal repeats providing a proxy for increasing age of each element [38,39]. Structurally annotated LTR elements that were private to B73 relative to the other NAM genomes have substantially more very young (LTRs are 100% identical) relative to core and near-core LTR elements suggesting that at least a subset of private elements may represent relatively recent insertions (Fig 2C).

## Different types of TEs exhibit variable polymorphic frequencies

We proceeded to investigate the relative frequencies of polymorphic TEs for different superfamilies of transposable elements. TEs can be subdivided based on their annotation method (structural *vs* homology) as well as their superfamily. There are four major superfamilies of class I elements—LINE (RIL), LTR-*Copia* (RLC), LTR-*Ty3* (RLG), and LTR-Unknown (RLX) —and five superfamilies of class II elements—DTA, DTC, DTH, DTM, and DTT based on homology to the *hAT*, *CACTA*, *Pif/Harbinger*, *Mutator*, or *Tc1/Mariner* superfamilies respectively. The retrotransposon superfamilies (RLC, RLG, and RLX) did not show substantial variation in the proportion of elements that were classified as polymorphic and had similar frequencies for both homology and structural annotations. In contrast, class II TIR transposon superfamilies exhibited more variable frequencies of polymorphic TEs (Fig 3A). Both homology and structural annotations of DTT elements exhibited relatively low (<20%) frequencies of polymorphic calls. The structural annotations of the other superfamilies were much more likely to be polymorphic with nearly 50% of structural DTA elements classified as polymorphic. We encourage exercising caution when drawing conclusions from differences in polymorphic TE proportions. It is unclear whether these differences were due to biological differences among superfamilies or technical artifacts such as differences in sizes or annotations.

TE superfamilies can be further subdivided into families, which contain related elements. In order to assess whether there were specific TE families with high levels of polymorphic elements, we also did a per-family analysis of the frequency of polymorphic elements. Each TE family was assessed in 25 total contrasts between B73 and each NAM genome providing multiple estimates of the frequency of polymorphic copies within a TE family. The range of the percent of polymorphic elements for a given family provides insight into whether that family had particularly high or low frequencies of polymorphic elements in one or a handful of contrasts. We limited this analysis to families with at least 20 members in B73 (N = 2,939 families). We found substantial variability in the proportion of polymorphic elements for small families (Fig 3B). However, larger families (> 1,000 copies, N = 168) tended to have a constrained proportion of polymorphic TEs, capped at 20% polymorphic (Fig 3C). In addition, there was

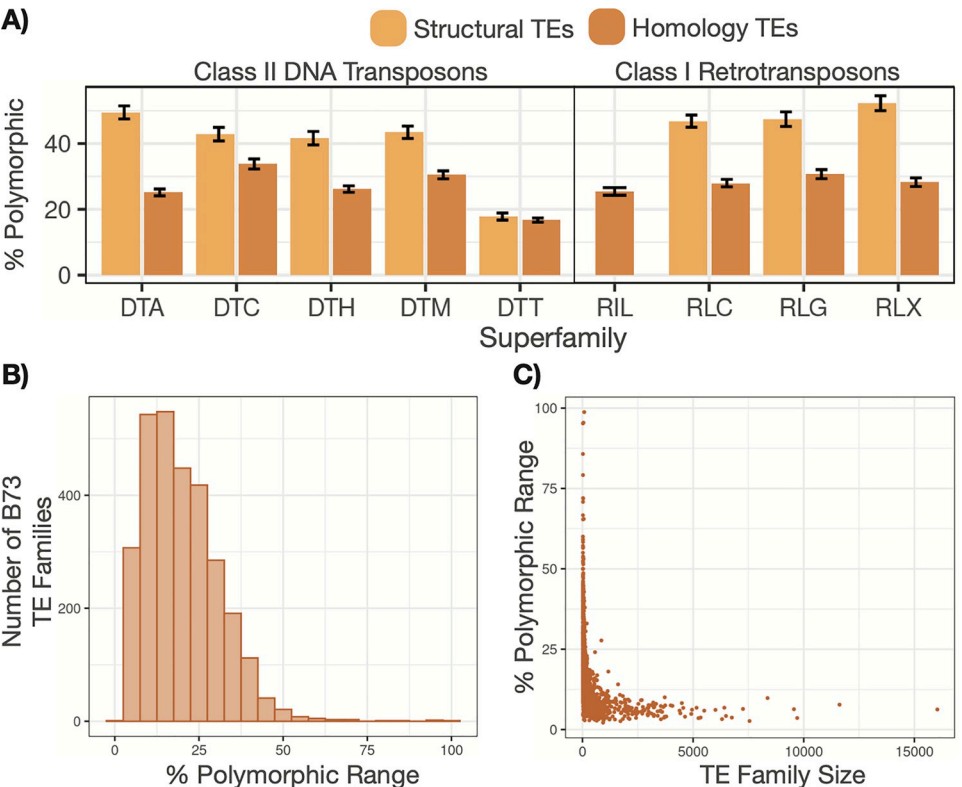

**Fig 3. Distribution of polymorphic B73 TEs across superfamilies and within distinct families.** (A) B73 TEs were grouped both by their assigned superfamily (x-axis) and by whether they were a structurally-annotated (yellow) or homology-annotated (orange) element. There are no structurally annotated RIL elements. Within a specific B73 versus NAM comparison, the percent of those elements classified as polymorphic was calculated. The height of the bar shows the mean percent across all 25 comparisons with error bars showing the standard deviation. (B) For each B73 TE family with greater than 20 copies in the genome, we determined what percent of that family was classified as polymorphic in a given comparison. We then plotted a histogram showing the range of these percent polymorphic values across all 25 comparisons. (C) The x-axis shows the number of copies for B73 TE family (> 20) in the B73 genome, while the y-axis shows the percept polymorphic range for that family.

limited evidence for outlier genomes in these contrasts (Fig 3B–3C). This suggested that there were few examples of specific TE families that had increased substantially in copy number in B73 relative to any of the other genotypes or vice versa.

## Polymorphic TEs are often located within larger structural variants

There are many thousands of polymorphic TEs identified in any genome comparison in maize. It can be tempting to think of these as representing simple TE insertion polymorphisms. However, visualization of specific chromosomal regions revealed that in many cases a single structural variant (SV) included multiple TEs from distinct superfamilies as well as non-TE sequence (Fig 4A and 4B). A careful examination of one of these SVs that was present in 14 of the NAM genomes reveals two TE fragments that are polymorphic and two TEs at the edges that were ambiguous due to the boundaries of this structural variant falling within annotated TEs (Fig 4A and 4B). These likely represented deletion events that removed TEs present in the ancestral sequence. A careful analysis of the boundaries of this SV reveals that all 14 genotypes have nearly identical boundaries for the SV with 2 of the 14 having boundaries that are shifted 1 base pair relative to the other 12. These boundaries did not reveal evidence for target site

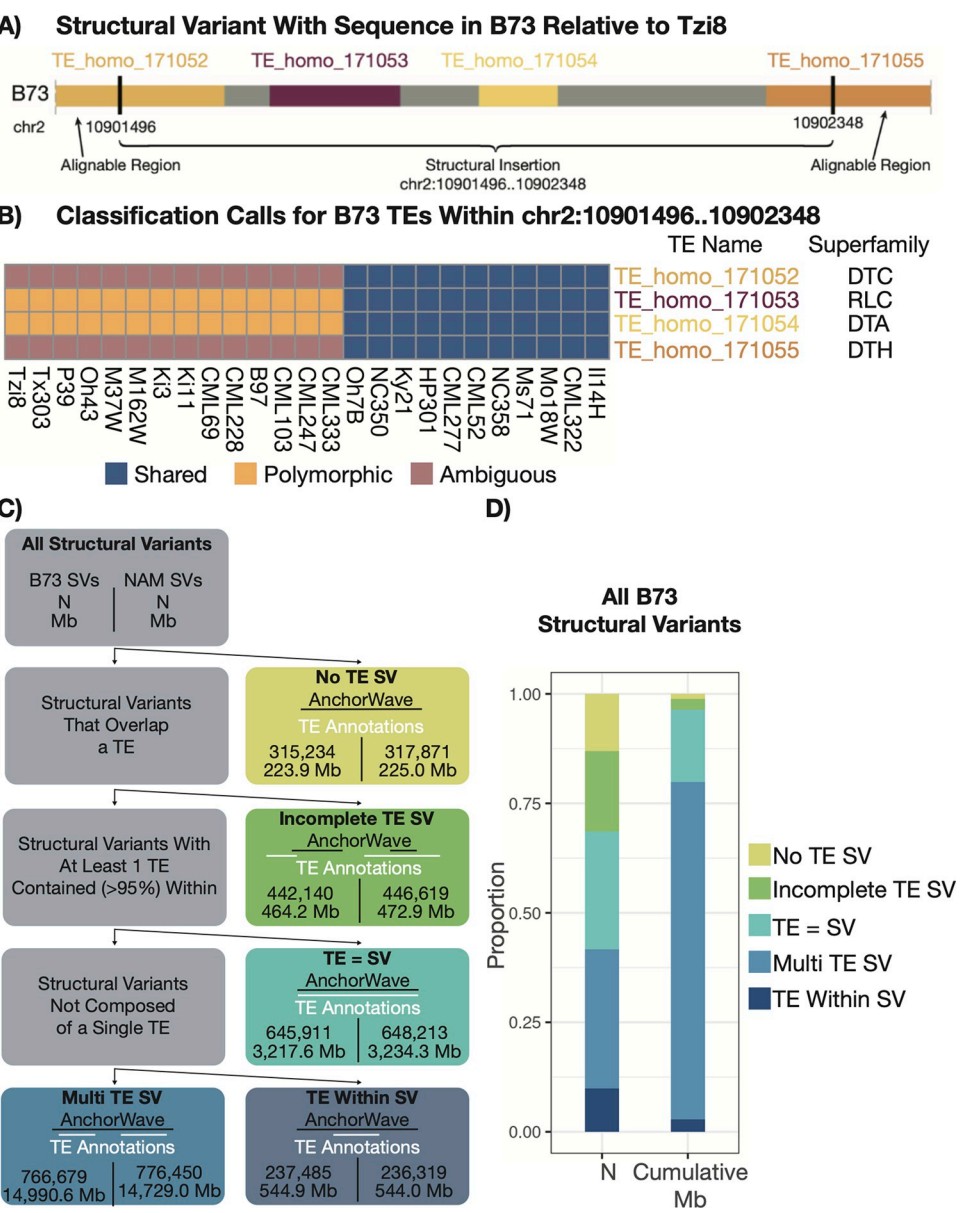

**Fig 4. Structural Variants Exhibit Differential TE Content.** (A) A visualization of a SV sequence in B73 relative to Tzi8 that is flanked by alignable regions. TE annotations in this region are overlaid with different colors representing the superfamily of the TE. (B) A heatmap showing the classification calls for the B73 TEs present in the SV shown in (A) across all of the NAM genomes. Colors represent whether the TE was classified as shared (blue), polymorphic (yellow), or ambiguous (purple) in that comparison. Columns are clustered based on similarity of alignable proportions of visualized TEs and show two distinct haplotypes where the SV is present (left) or absent (right). (C) A flowchart showing how SVs were classified depending on their overlapping TE content. (D) A stacked bar chart showing the proportion of SVs (in terms of total number–left column–and cumulative Mb–right column) with sequence in B73 relative to another NAM genome with colors delineating the different classes of SVs characterized in (C).

duplications or extended homology that would suggest a recombination event. Instead of solely focusing on a TE-centric analysis of variation as described above, we proceeded to characterize the full set of large (> 50bp) structural variants to understand how these SVs may be associated with transposon sequence(s) (Fig 4C).

In the analysis of all NAM genomes compared to B73, there were a total of > 2.4 million SVs (> 50bp) that represented sequence present in B73 absent in a NAM comparison that cumulatively accounts for 19,441 Mb of sequence (Fig 4C and 4D). These exhibited a wide spectrum of variation for the size of the SV sequence with 54.7% that were > 1kb, 22.7% that were greater than >10kb, and 0.43% that were greater than >100kb. An analysis of the TE annotations within the SV sequences revealed that 13% of the SV sequences in B73 did not overlap any annotated TE sequence, and these SVs are classified as 'No TE SV'. These 'No TE SV's tended to be relatively small (min = 51bp, median = 155bp, max = 340,969bp) and only accounted for 1.2% of the total Mb of SV sequence in B73. There were another 18.4% of SV sequences (2.4% of SV sequence Mbs) that only have partially (<95%) overlapping TE sequences (*i.e.*, all TE annotations that overlap the SV have <95% of the TE sequence overlapping the SV), which were classified as 'Incomplete TE SV'. These included examples in which only one edge of the TE overlapped the SV sequence as well as instances of SV sequences that were located entirely within a longer TE sequence. These 'Incomplete TE SV' overlaps likely represent deletion events that removed either one edge of the TE or an internal portion of the TE in the non-B73 genome.

The remaining 68% of SV sequences included at least one TE that was > 95% contained within the SV. These were divided into three further groups. The 'TE = SV' group was limited to instances in which both > 95% of the TE overlapped the SV and > 95% of the SV overlapped the TE suggesting a simple connection between a single TE and an SV. These accounted for 26.8% of SVs and 16.6% of the total Mbs of SVs. These 'TE = SV's likely represent TE insertion polymorphisms as well as some potential TE excision events for DNA transposons. The other two groups that contained at least one full (> 95%)TE were divided based on whether they included multiple full length TEs ('Multi TE SVs') or only a single full TE that accounted for less than 95% of the SV sequence ('TE Within SV'). The 'Multi TE SV's made up 31.8% of SVs and accounted for 77.1% of the Mb of SV sequence, of which 84% is actually TE sequence. These 'Multi TE SV's could be the result of a deletion that removes multiple adjacent TEs or could reflect a set of nested TEs that occur due to insertions of TEs within another TE. The 'TE within SV' category accounted for the remaining 9.8% of SVs and 2.8% of the Mbs of SV sequence. 'TE within SV's could be the result of an incomplete annotation of an older TE with some decay of structural features or could be a deletion event that removed a TE as well as additional sequences beyond the TE. The same analyses were also performed for the SVs that are present in the NAM genomes and absent in B73 and revealed similar trends as B73 present sequences (Fig 4C).

Based on the TE annotations used in this study, 77% of the maize genome is annotated as TEs. The classification of SVs relative to TE annotations revealed that, on average, 85% of the total SV sequence in B73 compared to any one NAM genome is annotated as TE, reflecting only a slight enrichment of TEs within SV sequence. However, while TE sequence was a major component of SV sequences, our analyses suggest that simple 'TE = SV' polymorphisms only account for a fraction of the SVs. Our TE centric comparison of genomes identified a total of approximately 6.47 million B73 polymorphic TE classifications across all comparisons of B73 to the other NAM genomes with a mean of 258,926 B73 polymorphic TEs in a given comparison to any one of the NAM genomes. The vast majority (99.9%) of these polymorphic TEs had >95% overlap with a single SV and these polymorphic TEs could be assigned to the 'TE = SV' (671,219–10.4%), 'Multi TE SV' (5,556,116–85.9%), and 'TE Within SV' (237,432–3.7%) groups. Overall, these analyses suggested that the majority of polymorphic TEs were not necessarily due to simple SVs that correspond precisely to a single TE but were instead due in large part to more complex SVs that include either multiple TEs or non-TE related sequences.

## Polymorphic TEs within SNP depleted blocks

The genome-wide comparisons of SVs and TEs suggested many complex contributions of TEs to structural variation beyond simple insertion polymorphisms. However, the extended divergence time of maize haplotypes allows for the potential of sequential insertions that result in nested TE annotations that could result in complex 'Multi TE SV' classifications. In order to better assess the relative frequency of putative insertion and deletion events, we decided to focus on recently diverged haplotypes that could provide insights into ongoing insertion and deletion events in the maize genome.

Genomic regions that were highly alignable with very low SNP rates when comparing B73 to another NAM genome likely reflected chromosomal regions recently derived from a common ancestor. In fact, relatively long (> 1Mb) SNP depleted regions are likely diverged for only tens to hundreds of generations. Using the pairwise AnchorWave alignments, we sought to identify these large SNP depleted regions in order to use them to monitor relatively recent changes in TE content in the NAM genomes (S3 Fig). Our analysis was restricted to regions of at least 2Mb with a SNP rate that was at least 100-fold lower than the genome-wide average SNP rate (see methods for details). There were a total of 213 of these SNP depleted regions that were identified based on comparisons of B73 to all of the other 25 NAM inbred parents (S2 Table). The size of the regions was quite variable (min = 2Mb, median = 3.1Mb, max = 25.5Mb) (S4 Fig). While many (26 out of 213) of the regions were only 2Mb in length in B73, there were also 23 SNP depleted regions that were at least 10Mb in length. These 213 SNP depleted regions accounted for approximately 1,048Mb cumulative base pairs of the B73 genome. SNP depleted regions were found across all ten chromosomes. There were no SNP depleted regions identified in comparisons of B73 and four of the NAM genomes (CML69, CML247, CML277, and NC350). In contrast, > 20 regions were identified between B73 and each of B97 (N = 22), Ky21 (N = 27), MS71 (N = 21), Oh7B (N = 25), and Oh43 (N = 31).

We expected to observe little structural variation across all SNP depleted regions given the relatively short divergence time between the two pairs of haplotypes. The percent of Mb attributable to structural variant sequences in SNP depleted regions was greatly reduced in comparison to the genome wide percentage (Fig 5A), and we found highly similar TE content as well. The B73 haplotypes in these regions contained 424,665 TEs (including 31,230 structurally annotated TEs). The vast majority (> 99.9%) of these TEs were classified as shared (Fig 5B). There were 410 B73 TEs in these regions classified as polymorphic and another 110 TEs classified as ambiguous. A similar analysis of the 424,577 TEs annotated in the NAM haplotype sequences for these regions revealed 576 polymorphic and 127 ambiguous TEs. The 986 polymorphic TEs within the SNP depleted regions likely represented a combination of novel insertions as well as deletion and/or excision events that could remove TE sequences. The analysis of SVs within these regions identified a total of 690 SVs including 139 'TE = SV' putative insertions in B73 and 79 'TE = SV' putative insertions in the non-B73 genomes (Fig 5C and 5D).

The majority (78.7%) of the 986 polymorphic TEs were located within 'Multi TE SV' (725 TEs in 126 SVs) or 'TE Within SV' (37 TEs in 37 SVs) variants and likely represented recent deletions that have removed TEs since it is unlikely to generate novel nested insertions or have poorly annotated young TE insertions in these recently diverged SNP depleted regions. There are also 179 'Incomplete TE SV' events in which a putative deletion results in the loss of an internal portion or one edge of an annotated TE. Together, these represent 342 SV events that are more likely due to recent deletions with 218 'TE = SV' events that might be more likely to be recent insertion events. This suggests a bias towards more deletion SV events than insertion SV events. The SV deletion events also account for substantially more total sequence, 1.928Mb relative to insertions which account for 1.14Mb.

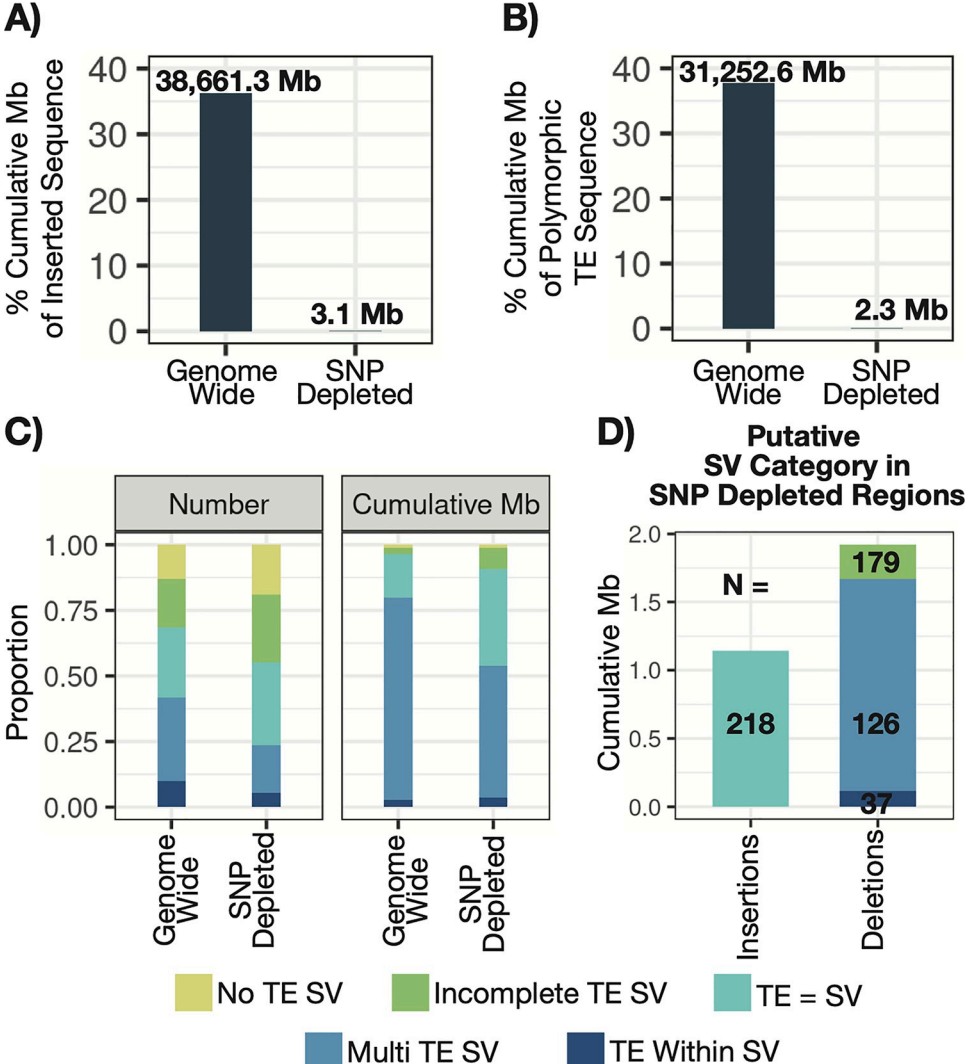

**Fig 5. Structural and TE Variation in SNP Depleted Regions.** (A) The percent cumulative Mb of inserted sequence from structural variants combined across B73 and NAM comparisons both genome-wide (left) and in SNP depleted regions (right) out of the total cumulative Mb of genomic sequence. (B) The percent cumulative Mb of polymorphic TE sequence present in B73 and NAM comparisons both genome-wide (left) and in SNP depleted regions (right) out of the total cumulative Mb of TE sequence. (C) A stacked barchart showing the proportion of structural variants (in terms of total number–left panel–and cumulative Mb–right panel) with sequence in one genotype relative to another across our B73 to NAM comparisons with colors delineating the different classes of SVs. The proportion is shown both genome wide (left) and in SNP depleted regions (right). (D) SV groupings can be further reduced into either putative insertions (left), comprised of 'TE = SV' events, or putative deletions (right), comprised of 'Incomplete TE SV', 'Multi TE SV', and 'TE Within SV' events. A barchart shows the cumulative Mb of these putative SV categories in SNP depleted regions.

The 218 'TE = SV' polymorphic TEs were candidates for relatively recent insertion polymorphisms from active TE families. These 224 TEs were from many different superfamilies of TEs and had relatively balanced numbers of insertions in the NAM genomes with no evidence for massive burst in any particular genome (S5 Fig and S3 Table). A per-family analysis revealed four TE families that account for > 10 'TE = SV' polymorphisms within the SNP depleted regions, and, in total, these four families accounted for nearly 60% of all 'TE = SV' events in these SNP depleted regions (Fig 6A). These include 67 members of the family

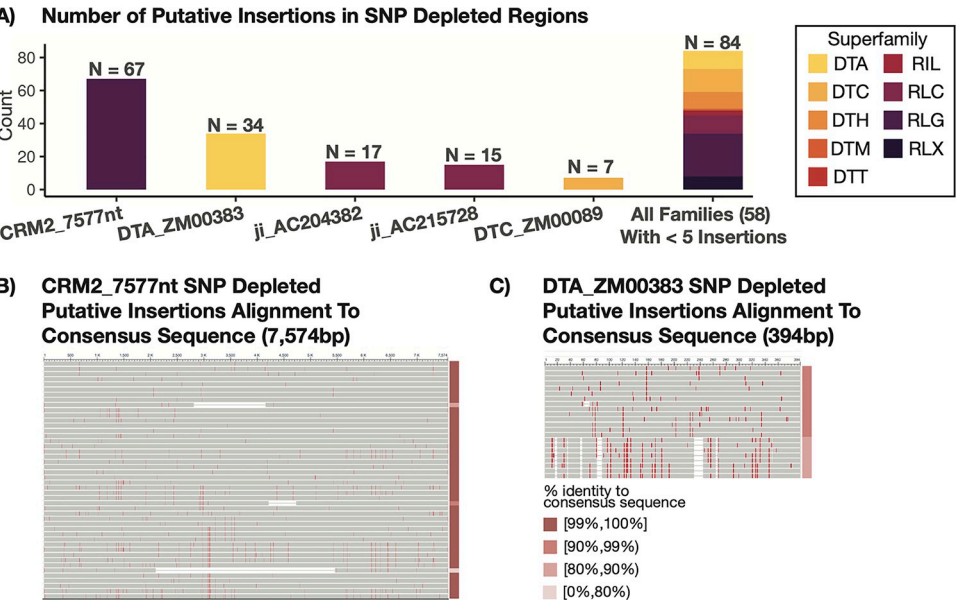

**Fig 6. Putative Insertions in SNP Depleted Regions Highlight Potentially Active TE Families.** (A) Putative insertions in SNP depleted regions were grouped by family to assess whether specific families show higher insertion events, suggesting potential transposition activity. A barchart shows the number of putative insertions for the five TE families with greater than five insertion events in SNP depleted regions, as well as a stacked bar showing the cumulative count of the remaining 58 families all of which have less than five insertion events. The specific number of events is annotated above each bar. Color indicates the superfamily classification for that TE family. Putative insertions (excluding solo elements) in SNP depleted regions belonging to the TE family CRM2_7577nt (B) and DTA_ZM00383 (C) were aligned to the consensus sequence for that family used in the annotation process. Vertical red lines indicate the presence of SNP relative to the consensus sequence. The percent identity to the consensus sequence is represented using color in the bar to the right of the multisequence alignment with darker pink colors representing a greater percent identity to the consensus sequence.

CRM2_7577nt, 34 members of the family DTA_ZM00383, 17 members of the family ji_AC204382, and 15 members of the family ji_AC215728. We characterized several features about these families with > 10 polymorphic 'TE = SV' events. In each case, there are roughly similar numbers of insertions in B73 and each of the other NAM genomes, and we found evidence for insertions in multiple genomic regions (S6 Fig).

Each of these families exhibited evidence that a specific subset of elements might be active. For example, the majority of the CRM2_7577nt insertions (61 of 67) were between 7500bp and 7600bp with ten insertions each that were exactly 7577 or 7581 base pairs in length (S6 Fig). Similarly, all putative insertions in SNP depleted regions belonging to the family DTA_ZM00383 were between 395bp and 406bp with 21 of 34 members being exactly 395bp in length (S6 Fig). At a per-family level, sequence identity between the putative insertions in SNP depleted regions to the consensus sequence used in the annotation process was determined (Figs 6B, 6C and S7). These putative insertions show high sequence similarity across all four families. With the exceptions of TE sequences with small InDels present, putative insertions in SNP depleted regions for CRM2_7577nt all show greater than 99% identity to the consensus sequence (Fig 6B). Putative insertions for DTA_ZM00383 in SNP depleted regions similarly showed high sequence similarity to the consensus, and SNP content amongst these insertions suggested two to three elements may have been actively proliferating (Fig 6C). Genome-wide analysis of the members of these families revealed substantially more variation in the size of all elements in these families which suggests that only a subset of these elements might be capable of movement.

While the analysis of the SNP-depleted regions provided an opportunity to capture evidence for recent transposition events, it is likely that recent events were occurring genome-wide. The analysis of all 'TE = SV' polymorphic TEs revealed that these four families each have high levels of 'TE = SV' events. There are 8,307, 6,945, 11,481, and 17,599 events respectively for the families of CRM2_7577nt, DTA_ZM00383, ji_AC204382, and ji_AC215728. Among the 779 TE families with an average of > 100 copies per genome, these four families are all in the top 25% for proportion of 'TE = SV' events.

## Discussion

As researchers have moved from the era of single reference genomes for any species to producing multiple high-copy genome assemblies for genetically diverse individuals of a species, substantial variation in genome content has been observed [26,40]. We were particularly interested in studying the contribution of transposable elements (TEs) to this genomic variation. We sought to understand both the contribution of TEs to structural variants and to understand the relative dynamics of TE insertions compared to deletion events that might contract genome size *via* removal of TEs and other sequences.

The high-quality NAM genomes provide an opportunity to study these dynamics [26]. The TEs within these genomes were all annotated using a consistent approach [34]. From these annotations, we were able to classify the shared or polymorphic status for the majority of the elements in the genome. These classifications relied upon the precise whole-genome alignments generated by AnchorWave [33]. We found that there are abundant polymorphic TEs in the comparison of maize genomes. There are approximately 900,000 annotated TEs in any NAM genome. Across our comparisons, we found that ~650,000 of these TEs are shared between B73 and any NAM genome while the remaining ~250,000 TEs are polymorphic. These results are quite similar to prior work that assessed shared and polymorphic TE content in four genomes using a different approach for annotation and classification [3,29,41] or that assessed TE variation at specific genomic regions [31,32]. This leads to substantial variation among the genomes in terms of the presence of TEs at specific genomic regions and results in many genes being located near polymorphic TEs. Only a subset of maize TEs are shared among all the NAM genomes leading to highly variable haplotype structures.

While it can be quite useful to classify TEs as shared or polymorphic, the actual haplotype structures reveal more complex patterns. Often polymorphic TEs are part of a larger structural variant that includes multiple TEs. This can complicate functional analyses as it becomes impossible to monitor the potential effect of just one polymorphic TE as there are multiple TEs all in complete linkage. We therefore implemented a strategy to classify all structural variants (InDels > 50bp) in relation to TE annotations. This revealed that many (85.8%) of the polymorphic TEs in B73 actually occur as part of 'Multi TE SV' events. Only a subset (10.4%) of these polymorphic TEs are instances in which the boundaries of the TE and the SV are quite similar ('TE = SV' events). It would be quite interesting to perform a genome-wide classification of putative deletions and insertions based on the relative annotations of the TEs and SVs. However, the long divergence time for many maize haplotypes can confound this analysis due to combinations of nested TE insertions and mutations that reduce the accuracy of annotation of precise TE boundaries. Genomic regions with less divergence time can provide insights into more recent insertion / deletion dynamics.

Maize is an outcrossing species and several of the NAM inbreds have pedigrees that include common parents. This leads to the presence of large SNP depleted regions that likely represent inheritance of an extended haplotype from a common parent in the past 10s to 100s of generations. These regions provide an opportunity to study the ongoing dynamics of structural

variation and TE polymorphisms in modern maize inbreds. While these regions have very few SNPs as expected for common descent, there are still some examples of structural variants and TE polymorphisms. It is worth noting that >99.9% of the annotated TEs within these regions are classified as shared suggesting that the ongoing movement of TEs is quite limited. However, the rare polymorphisms within these regions provide insights into the current expansion and contraction of TEs. Only 218 (23%) of the 986 polymorphic TEs in SNP depleted regions represent 'TE = SV' events. The majority of the polymorphic TEs likely occur as a result of deletion events rather than novel insertions. In these regions, there are a total of 1.93 Mb of putative deletions ('Incomplete TE SV', 'Multi TE SV', and 'TE Within SV') and 1.14 Mb of putative TE insertions ('TE = SV' events). This suggests that the maize genome may be in a phase of contraction rather than expansion as TE removal is currently outpacing TE insertions.

The characterization of the 'TE = SV' events within these SNP depleted regions identified several TE families with potential transposition activity in modern germplasm. These included several LTR families (CRM2_7577nt, ji_AC204382, and ji_AC215728) and a DNA TIR family (DTA_ZM00383). We did not find evidence that these families were only active in particular genotypes but instead see likely evidence for low rates of movement across multiple genotypes. There do seem to be particular members of these families that are mobile as many of the putative insertions are of quite similar length and sequence identity. Genome-wide analyses suggest high levels of polymorphic TEs within these families. Further studies to characterize these potentially active TEs will be important in characterizing potential autonomous elements and the impacts of these TEs.

Combined our results highlight how both insertions and deletions of transposable elements contribute to structural variation in maize. It is compelling to think of TEs as generating insertions. Both their mechanisms of movement and characterization as 'selfish' genetic elements evoke the idea of sequence gain. However, unmanaged TE proliferation would lead to genomic bloat as genomes become riddled with TEs. While genomes have evolved myriad mechanisms to silence TE proliferation, ongoing deletion events provide another strategy for mitigating the spread of TEs. Genome-wide deletion rates are likely governed by factors independent of TE transposition rates, but TE proliferation could facilitate ectopic recombination events between similar TE sequences, resulting in a deletion event. Further investigation into the insertion and deletion rates of TEs is warranted to better understand how TEs influence genomic variation.

## Methods

### Characterization of structural variation among NAM genomes

High-quality genomic sequences have been produced for the 26 NAM inbred founder lines [26]. AnchorWave v1.0.1, a recently developed approach used to perform pairwise whole-genome alignments, was used to compare each of the NAM inbred genomes to B73 [33] via the 'genoAli' command and '-IV' parameter. The MAFToGVCF plugin of tassel v5.2.82 [42] was used to reformat genome alignments in MAF format into variant calling records in GVCF format. The resulting GVCF files contain records of all nonvariant and variant sites, including single nucleotide and structural differences. Given our interest in shared or polymorphic TEs, we condensed the GVCF output by combining the nonvariant sites, single nucleotide variants, and small (<50bp) insertion/deletion variants into a single class of 'alignable' regions. 'Structural variants' in one genotype relative to another were defined as regions > 50bp for which the size in the other genotype is 0bp. The remaining variants all include at least one base pair in each genotype that is not fully aligned, and these regions were consequently classified as 'unalignable'. Gaps in the AnchorWave alignment were identified and classified as 'missing data', but these regions were treated as unalignable in all downstream analyses.

## TE annotation processing

The TE content for each of the NAM genomes had been previously annotated using panEDTA [14,25,26,34,35]. These annotations are publicly available for download through MaizeGDB (https://maizegdb.org/NAM_project). This process initially identifies TEs based on structural features using tools including LTR_Finder [43], LTRharvest [44], and TIR-Learner [45]. These structurally identified TEs are then used to create a panTE library across all of the genomes that is used to perform homology based annotation of non-structurally intact TE fragments. We filtered out annotations for non-TE repeats, helitrons (due to lower annotation quality for these elements [35], and specific features of structurally annotated LTRs (e.g., target site duplications, long terminal repeats) such that we only retain the full-length structural annotation that includes the LTR regions. We removed a small percentage of 'duplicate' annotations with different IDs but identical coordinate positions, superfamily classifications, and family classifications (median of N = 418 annotations, ~0.037%, removed per NAM line). With duplicate annotations, we prioritized retaining structural annotations over homology annotations. If both annotations were identified using the same method, we randomly decided which one to keep.

Preliminary analyses of the previously released output using bedtools (v2.30.0) [46] identified potentially problematic overlapping annotations. One family ('DTA_ZM00081_consensus') frequently had homology annotations that occurred in multiple regions throughout LTR elements suggesting potential contamination of this TIR element with LTR related sequences in the original Maize TE Consortium (MTEC) library that carried forward into the panEDTA library. Therefore, all annotations with this family ID were removed from downstream analyses. We also identified examples of overlapping TE annotations that seemed biologically unfeasible. TE annotations whose start or end position was within 5 base pairs of the start position or within 5 base pairs of the end position of a structurally annotated TE were filtered out. If two structural annotations overlapped in this way, we prioritized retention of the larger structural annotation and randomly selected one for retention if they were equally sized. Any homology TE annotation that had greater than 10% overlap (including those that were contained with 100% overlap) with another homology annotation was removed. Additionally, any homology TE annotation that overlapped a structural TE annotation by greater than 5% but was not fully contained within that structural annotation was removed. Finally, we filtered out structural TE annotations that overlapped but were not contained within another structural TE annotation. We prioritized retention of Class II annotations and longer annotated elements when deciding which structural annotation to keep. This resulted in a final annotation in which there are very few examples of the same region being annotated as part of multiple TE features and allows for a more accurate assessment of the TE base pairs within each of the genomes.

## Gene annotation processing

Annotated genes for each NAM line were characterized and similarly obtained from Hufford et al. (2021) [26]. Annotations could be further broken down into exon-only or full-length annotations. As part of this work, each gene was classified as either syntenic or non-syntenic relative to sorghum. A full description of how synteny assignments were determined can be found in the Supplementary Materials (Fractionation Analysis) of Hufford et al. 2021 [26]. In summary, filtered exons from the outgroup Sorghum bicolor (Sbicolor_313_v3.1 from Phytozome) were aligned to the repeat masked NAM and B73 genomes. Orthologs were scored in each NAM line based on alignment of at least one Sorghum exon to a single gene-space locus syntenic with the query Sorghum gene. Orthologs could be classified as either fully retained,

partial deletions, or fully fractionated. These results were then filtered for Sorghum exon alignments falling within the identified subgenome blocks of B73 version 4 associated with syntenic coordinates of Sorghum gene models. Gene density for each subgenome was determined by aligning the primary Sorghum CDS gene model sequences to the primary CDS gene model sequences of each NAM line. Syntenic relations were then determined from these alignments.

## Identification of shared and polymorphic genomic features

The processed AnchorWave GVCF files allowed classification of all segments of pairwise contrasted genomes into alignable, inserted, or unalignable sequence. The genome-wide annotations for transposable elements or genes were then intersected with these regions using bedtools (v2.30.0). Any features that had at least 95% of its sequence overlapping alignable regions was classified as a shared TE between the contrasted genomes. Features were classified as polymorphic if they had at least 95% sequence overlap with a structural variant in their own lineage (*i.e.*, B73 TEs were polymorphic if they had 95% overlap with structural variants with sequence in B73). The remaining features that did not include at least 95% overlap with either alignable or structural variant sequence were classified as ambiguous, as their exact status (*i.e.*, shared or polymorphic) could not be confidently determined.

We classified both the filtered EDTA TE annotations and the canonical gene annotations across the NAM genomes as either shared, polymorphic, or ambiguous. Canonical gene annotations were classified using either exon-only coordinates or full-length coordinates for each model. Due to the pairwise nature of the AnchorWave alignments, the B73 TE annotations were classified 25 times, one for each query comparison, while the TE annotations for the remaining 25 NAM genomes were only classified once in relation to their presence or absence in B73.

## Identification of SNP depleted regions

Between B73 and each of the NAM genomes, the number of SNPs and amount of alignable sequence called using the AnchorWave alignment was used to identify SNP depleted regions in each pairwise comparison. We used a sliding window approach to count the number of SNPs and base pairs of alignable sequence in 1Mb windows offset by 250kb from the start to end of each chromosome. Normalized SNP counts for each 1Mb window were then determined by dividing the SNP count by the total amount of alignable sequence in the 1Mb window. We identified the subset of 1Mb windows that had >950,000 base pairs of alignable sequence and had a normalized SNP rate less than 1 in 10,000 (the average SNP rate was 1 in 44). We further required a minimum of 5 consecutive 1Mb sliding windows that meet these criteria in order to identify at least 2Mb regions in pairwise comparisons that were highly depleted of SNPs and likely represent identity by state. For all analyses, we offset the start and end coordinates for each SNP depleted region by 100,000 base pairs to ensure the boundaries of the region did not extend beyond the putative identity by state region of the genome.

## Evaluation relatedness of putative insertions in SNP depleted regions

Four TE families had several putative insertions within SNP depleted regions–CRM2_7577nt, DTA_ZM00383, ji_AC2043821, and ji_AC215728. Within each family of interest, all TE sequences identified as putative insertions in SNP depleted regions across all of the pairwise comparisons were extracted. Any solo elements were dropped from further analysis. To determine the phylogenetic relationship between copies within each family, the remaining sequences were aligned with MUSCLE using default settings [47]. These aligned sequences were then trimmed using trimAL with parameters '-automated1' [48]. Trimmed sequences

were then aligned again using MUSCLE and default parameters. A phylogenetic tree was generated for eacfh family using RAxML with settings '-f a -m GTRGAMMA -p 12345 -x 12345 -# autoMRE' [49]. Phylogenetic trees were plotted with the ggtree R package [50].

## Dryad DOI

https://doi.org/doi:10.5061/dryad.5qfttdz9t [51]

## Supporting information

**S1 Fig. Proportions of AnchorWave Classifications Across NAM Lines.** For each pairwise contrast between B73 and a NAM genome, each region of the pairwise AnchorWave alignment could be classified as either alignable (dark blue), structural variant sequence (orange), or unalignable (purple). (A) Barplots showing the proportion in Mb of the B73 genome classified as each group against every NAM line (x-axis). (B) Barplots showing the proportion in Mb of every NAM genome (x-axis) classified as each group against B73. The last bar shows the average across all NAM lines with the specific percentages listed to the right of them. (PNG)

**S2 Fig. Proportions of Feature Classifications Across All Pairwise Comparisons.** TE annotations (A,B), exon-only gene annotations (C,D), and full-length gene annotations (E,F) were intersected with pairwise AnchorWave alignments to classify each feature as either shared, (light blue), polymorphic (yellow), or ambiguous (light purple). Proportions of B73 features against every NAM line (A,C,E) and proportions of every NAM feature against B73 (B,D,F) classified as shared, polymorphic, or ambiguous. The last bar shows the average across all comparisons with the specific percentages listed to the right of them. (PNG)

**S3 Fig. Visualization Highlighting Identified SNP Depleted Regions From the B73 vs B97 AnchorWave Pairwise Alignments.** Each panel shows a different chromosome with the x-axis indicating the start position of a 1Mb window with the y-axis showing the normalized SNP count (raw SNP count/base pairs of alignable sequence). Yellow boxes indicate regions identified as SNP depleted exhibiting a normalized SNP rate less than 1 in 10,000. Gaps represent windows of either missing data or regions where there was no alignable sequence. (PNG)

**S4 Fig. Distribution of Sizes for SNP Depleted Regions Across AnchorWave Pairwise Alignments.** The x-axis shows the size of the region in megabases, while they y-axis shows the count (PNG)

**S5 Fig. Distribution of Polymorphic TEs in SNP Depleted Regions.** The first row shows this distribution by superfamily, while the second row shows this distribution by chromosome. (A) and (C) partition polymorphic TEs based on whether they are present in B73 or if they are present in a NAM line. For polymorphic TEs in NAM lines, (B) and (D) expand on this to show the distribution across NAM lines. (PNG)

**S6 Fig. Distribution of Polymorphic TEs in SNP Depleted Regions for Potentially Active Families.** Four families were identified as potentially being active due to a high number of polymorphic TEs in SNP depleted regions: (A) CRM2_7577nt (N = 67), (B) DTA_ZM00383 (N = 34), (C) ji_AC204382 (N = 17), and (D) ji_AC215728 (N = 15). Some TEs were dropped to improve visualization including 6/67 polymorphic TEs for CRM2_7577nt and 1/17

polymorphic TEs for ji_AC204382. Column 1 in each plot shows the size distribution of polymorphic TEs in SNP depleted regions. Column 2 shows the number of polymorphic TEs in either B73 or NAM partitioned by chromosome. For polymorphic TEs present in NAM, column 3 shows the specific distribution across lines.
(PNG)

**S7 Fig. Sequence Identity Between Putative Insertions in SNP Depleted Regions for Top Families.** Four families were identified as potentially being active due to a high number of polymorphic TEs in SNP depleted regions: (A) CRM2_7577nt (N = 67), (B) DTA_ZM00383 (N = 34), (C) ji_AC204382 (N = 17), and (D) ji_AC215728 (N = 15). Some TEs were dropped to improve visualization including 6/67 polymorphic TEs for CRM2_7577nt and 1/17 polymorphic TEs for ji_AC204382. Each row represents an alignment to the consensus sequence for that family.
(PNG)

**S1 Table. Summary of Filtered TE Content Across NAM Lines.** Substantial filtering was done on the publicly available panEDTA TE annotations for each NAM line. The total number and Mb of TE sequence genome-wide, as well as by various features (including TE class, TE superfamily, TE annotation method), was listed for all NAM lines.
(CSV)

**S2 Table. Location of all SNP Depleted Regions.** The coordinates for all SNP depleted regions identified in our analysis as well as their size.
(CSV)

**S3 Table. Candidate TEs for Recent Insertion Polymorphisms.** 224 TEs contributed to 'TE = SV' polymorphic TEs in SNP depleted regions that were candidates for relatively recent insertion polymorphisms. Location of these TE annotations, structural variant block identified by AnchorWave, and the SNP depleted region they belong to were provided.
(CSV)

**S4 Table. Median Genomic Feature Contribution to AnchorWave Alignment Categories.** AnchorWave alignments were parsed into either alignable, structural variant, or unalignable sequence. These were intersected with TE annotations, syntenic genes, non-syntenic genes, and the remaining genome to discern the median contribution of these features to the three AnchorWave categories across our 25 pairwise alignments.
(CSV)

## Acknowledgments

We would like to thank Shujun Ou for sharing the panEDTA TE annotations with us directly. We would also like to thank Jeff Ross-Ibarra, Emily Josephs, Nathan Catlin, Zach Myers, Erika Magnusson, Cathy Rushworth, and Shelley Sianta for useful comments, insights, and suggestions throughout the analysis. Finally, we thank the Minnesota Supercomputing Institute at the University of Minnesota (https://www.msi.umn.edu) for providing resources that contributed to the research results reported within this article.

## Author Contributions

**Conceptualization:** Manisha Munasinghe, Nathan Springer.

**Data curation:** Manisha Munasinghe, Candice N. Hirsch.

**Formal analysis:** Manisha Munasinghe, Claire C. Menard, Kristy Yubo Ma.

**Funding acquisition:** Nathan Springer.

**Methodology:** Manisha Munasinghe, Andrew Read, Michelle C. Stitzer, Baoxing Song, Nathan Springer.

**Project administration:** Candice N. Hirsch, Nathan Springer.

**Resources:** Michelle C. Stitzer, Baoxing Song.

**Software:** Manisha Munasinghe, Michelle C. Stitzer, Baoxing Song.

**Supervision:** Yaniv Brandvain, Candice N. Hirsch, Nathan Springer.

**Visualization:** Manisha Munasinghe, Yaniv Brandvain, Nathan Springer.

**Writing – original draft:** Manisha Munasinghe, Nathan Springer.

**Writing – review & editing:** Manisha Munasinghe, Andrew Read, Michelle C. Stitzer, Baoxing Song, Claire C. Menard, Kristy Yubo Ma, Yaniv Brandvain, Candice N. Hirsch, Nathan Springer.

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
