## [Decision Letter · Decision Letter 0]

18 Jul 2023

Dear Dr Munashige,

Thank you very much for submitting your Research Article entitled 'Combined analysis of transposable elements and structural variation in maize genomes reveals genome contraction outpaces expansion' to PLOS Genetics.

The manuscript was fully evaluated at the editorial level and by independent peer reviewers. The reviewers appreciated the attention to an important problem, but raised some concerns about the capacity of the methodology to sustain the results found. Based on the reviews, we will not be able to accept this version of the manuscript, but we would be willing to review a revised version. We cannot, of course, promise publication at that time.

If you decide to revise the manuscript for further consideration at PLOS Genetics, please aim to resubmit within the next 60 days, unless it will take extra time to address the concerns of the reviewers, in which case we would appreciate an expected resubmission date by email to plosgenetics@plos.org.

We are sorry that we cannot be more positive about your manuscript at this stage. Please do not hesitate to contact us if you have any concerns or questions.

Yours sincerely,

Clémentine VITTE, PhD

Guest Editor

PLOS Genetics

Justin Fay

Section Editor

PLOS Genetics

Dear authors,

Thank you for submitting your work to PLOS Genetics. The manuscript from Munashige et al. contributes to the field by addressing a long-standing question on determining whether genome expanses brought by transposable elements (TEs) can be counter-balanced by other molecular processes. The rationale of the paper is clearly explained, and highly relevant for plant genome evolution.

The two reviewers point the interest of the biological question addressed as well as the adequation of the methodology set up to answer it. They also raise some specific points which are relevant to improve the manuscript.

In addition to the comments of the reviewers, I have myself some points that I would like the authors to address, in particular on the methodology used.

Although the overall rationale of the methodology (*i.e*., detecting structural variants between maize lines from whole genome sequence assembly comparisons, and crossing the results with TE positions from annotations) is relevant, I have concerns on the extent to which the combination of panEDTA and AnchorWave information allows to accurately answer the question addressed.

In particular, the authors observe a large number of structural variants (SVs) encompassing several TE copies, and conclude that these variants correspond to deletions. But couldn’t they also correspond to nested transposable elements structures, where TEs are piled into one another? If the oldest elements (at the bottom of the pile) are highly degraded, there is a possibility that a homology could be found on one side of the pile, but not on the other, leading to the prediction of two different elements at the two borders of the variant. This would lead to prediction of a deletion, where it is actually a block of insertions.

Nested cases like this have been described in Liu et al. 2007 – PNAS and checking how these cases are resolved by the methodology used is crucial to make sure that the methodology sustains the conclusions found.

More largely, a major point would be to determine whether the structural variants encompassing several TEs correspond to nested cases, or to series of independent insertions not piled within one another.

The authors provide a schematic view of the analysis of one region (Figure 1), but it is difficult to understand to what extent this example is representative of all cases. In particular, regions that have been previously characterized such as the *Bronze* region (Wang and Dooner, 2006 – PNAS) are not presented. Comparing the results found by the procedure used (positions of TE annotations predicted by panEDTA, position of SVs detected by AnchorWave) to these of previous analyses in well annotated regions is an important benchmark.

Also, to claim that the underlying mechanism is deletion, the authors should provide an analysis of the breakpoints. In particular, stretches of microhomology have been observed for presence/absence variants in several plant species including maize. Are similar patterns observed? What is the size range of the stretches and does it fit this observed in previous reports? (see for instance Darracq et al., 2014 – BMC Genomics; Vitte and Bennetzen, 2006 – PNAS)

All the analyses presented in Figure 3 are based on all types of TE variants. But these include both insertions of single elements, and variants including several TEs. How are the latest impacting the results? If the aim of this series of results is to compare the polymorphism of insertions among families and superfamilies, the analysis should be restricted to cases where TE=SV, without taking into account SVs encompassing several TE copies, which could correspond to other processes than insertion.

It would be nice to have a chromosomal representation of variants across the B73 chromosomes, to have an idea of local densities. A color code could be used to highlight the distribution of the different variant cases.

p.15, l.309-311: The large variants could also correspond to large insertions, rather than to large deletions. Could the authors state how they can discriminate the two?

p.16, l.316-319: Describe how many regions that corresponds to, what is there size distribution and where they are located in the genome. Move paragraph “The size of the regions…B73 genome” upper and give more details on size distribution and chromosomal locations.

The analysis of the SNP-depleted regions allows to find 139 cases of ‘TE=SV’ among a total of 690 cases (p.17, l.354). This makes about 20% of the cases. But in the rest of the genome, the authors find a percentage of about 10%. Could the authors comment on this? This suggests that non ‘TE=SV’ variants are more often occurring in the SNP-containing regions of the genome. Are SV sizes larger in the SNP-containing regions? This could fit with a larger abundance of nested TE in the more divergent regions, and could point to presence/absence of these nests rather than to large deletions.

As pointed reviewer #1, it is very important for the community to have access to all variants detected. This should be supplied as supplementary material.

There are also minor points that should be corrected in the manuscript:

p.2 l.32: Precise if referring to numbers of events, or number of base pairs

p.3 l.40: Add drift

p.4 l. 63: Actually, a large number of studies have analyzed the effect of deletions (see for instance Devos et aL;, 2022 - Genome Research and Vitte and Bennetzen 2006 – PNAS). The models did take into account the deletion process (see for instance Vitte and Panaud, 2005 – CGR). But due to limited quality of the genome assemblies at the time, these studies were limited to analysis of internal deletions located within TEs, and could not take into account large deletions. This is what is accessible now that high quality genome assemblies are available for several individuals.

p.4 l. 72. The authors should acknowledge that these pioneer studies gave important insights on TE-related maize genome structure and evolution, in particular on the importance of TE insertions and recombination to generate the TE landscape observed in different maize inbred lines. They were nevertheless limited in the amount of genomic sequence analyzed, and therefore whole genome assemblies now allow for accurate quantification of the processes involved.

p.6 l. 124: Move reference after ‘AnchorWave’.

p.6 l. 128: What is the range observed?

p.7 l. 147: Change ‘inserted sequence’ to ‘SV’

p.8 l. 157: Are the results in line with the genetic distance between B73 and the different NAM founders?

p.8 l.165: Change ‘low polymorphic rate’ to ‘low rate of polymorphic cases’

p.8 l.168: “… estimates”: add reference

p.9, l. 184: So what was the overlap between non syntenic genes and TEs in this study? Why were these ‘genes’ not considered as TEs?

p.9, l.188: “… TE mobilization”: add reference

p.10, l/214: Superfamilies full name should be indicated in italics

p.11 l.218-219: Isn’t this expected by the cut and paste mechanism?

p.11 l.229: Change ‘elements within a family’ by ‘copies within a TE family’

p.14, l.279: Please precise the size range observed

p.14, l.281: “SV sequences that have partially (>95%) overlapping TE sequences”: does this mean that 95% of the SV length is made of TE sequence or that 95% of the TE is within the SV? From the examples, it seems that it is the second case. Please clarify. Could cases of SV located within larger TEs correspond to misannotated insertion within the TE rather than to deletions? To what extent could such insertions be missed in the annotation?

p.15, l. 298: On average, 85% of the SVs is annotated as TE bases. This proportion is similar to this of TE sequences within the B73 genome. Could the authors comment on this? This could suggest that SV occur randomly in the genome rather than that TEs are major contributors to SVs.

p.17, l.334: How was the 100-fold threshold chosen?

p.17, l.339: Is this excepted considering the pedigree of the B73 inbred line? Where are these regions located? Are they shared between the NAM lines where they are found?

p.17, l. 343: ‘the SNP depleted regions’: do they correspond to the 213 or to the subset presented above?

p.18, l.356: why 224 and not 139?

p.18, l.365: ‘these regions’: please precise which ones

p.18, l.376: ‘sequence similarity’: do the authors mean ‘sequence identity’?

p.21, l.428: Please cite previous work, such as this of Wang and Dooner, 2006 - PNAS

p.22, l.454, l.457, l.458: please check typos and missing information

p.23, l.478: The authors could have proof of TE-related ectopic recombination using their data: what is the number of cases where the same TE family is found at the two borders, with breakpoint at the same site?

p.23, l.480: The authors should also discuss the presence of deletions within the TE sequences. What is their proportion in the analyzed dataset?

p.24, l.507: Were long terminal repeat regions really removed?

p.26, l.537: Please precise the fraction overlapping TE annotations

p.26, l.554-556: For SVs that were found between B73 and several maize lines, were the breakpoints found always the same at the base level? This would help deciphering the accuracy of breakpoint prediction by AnchorWave.

Figure 1:

Precise the amount of overlap required for ambiguous cases

Figure 1 B: Add colors showing the amount corresponding to the different features (TE, gene syntenic, gene non syntenic

Figure 3A: Please add a statistical test

Figure 3B: Please state how many families the >20 copies threshold leads to

Figure 3C: The pattern observed is also likely impacted by the age of the copies. How does the pattern correlate with age of the copies? Is the age of the small families similar to this of families with larger copy number?

Reviewer's Responses to Questions

**Comments to the Authors:**

Reviewer #1: In this work Munasinghe et colleagues analysed the independently assembled genome of several maize accessions. Interestingly, the analysis integrated structural variants and transposable elements (TEs) variations, allowing the integrated study of the plasticity of maize genome. Authors used regions with low SNPs accumulation to investigate recent structural variant events and managed to identify TE families with recent mobilisation in maize genome, likely occurred after domestication. This is a very interesting work which use a pan-genome comparison approach to reveal dynamics of maize genome plasticity. Authors concluded that many variants involving TE regions are likely events of deletions, while there is less evidence of TE new insertions. I believe these results are interesting and relevant to study how plant genomes evolve. One suggestion I have is related to the putative deletion events identified. Authors suggest that such variants could be the consequence of ectopic recombination events. So, I was wondering if authors have looked into a possible enrichment homology at the edges of these regions, which can be associated to non-homologous recombination events.

The remaining comments are quite minors, and are listed below:

Figure 1A. if B97 should be representative of any NAM line, please mention this in the legend, or use a general “NAM line” label in the figure panel.

Figure 3A. it looks RIL TEs miss the structural annotation bar. Is this intentional?

Line 101. Avoid nidified brackets

Line 162. The definition of syntenic and non-syntenic genes requires a better explanation. From method section, this appears to be done only in comparison to sorghum, but more information about the approach used is needed.

Line 268-269. Did authors check that edges of the regions do not contain TIRs, TSD or homology, which could support transposition (of a misannotated TE) or maybe a recombination event?

Line 298-300. Considering that maize genome contain >70% of transposons, this sentence is maybe excessive. I believe the finding that 85% of total SVs include TEs likely indicates an overrepresentation of TE DNA in variants, which if randomly distributed should be around 70% (nonetheless, a statistical test could be welcome here, especially considering that TEs are only associated to portion of “incomplete TE SVs”).

Line 334. I think the use of “>20” is misleading? Can the five specific number of region be given in full?

Line 341. Is 26 the number of regions or the reference to a citation? Can the precise number be given?

Line 454. Typo in “into”

457-458. what “X” stand for?

459-460. I believe an alternative possibility is that TE insertions have been underestimated (e.g. exclusion of Helitrons from the analysis).

Line 506. It is not explained why Helitrons (which are common TEs in maize genome) have been excluded from the analysis

Line 507. Typo in “duplications”

Line 508. Can you replace “handful” with a precise quantification?

The list of annotated TEs and structural variants should be made available as supplementary material. I believe the authors will also benefit in providing with their manuscript a .gff file containing the positions of the structural variants detected annotated on the B73 genome, as well as the TE annotation, because other researchers could directly use them in future genome comparison studies.

Reviewer #2: The manuscript submitted by Munasinghe et al. describes a comparative genomic analysis of several well assembled maize genome to study the TE-driven genomic differenciation in this crop species.

This paper is well written (apart from few lines in the discussion...) and the analyses are well described and I have no problem with most interpretations of the results.

The authors have used a whole genome alignment method (Anchorwave) combined with an annotation of TEs of the genome of 26 NAM lines (panEDTA) to identify Transposable element insertion polymorphisms (TIPs). Using this approach, they found that 30 % of all TEs are polymorphic between B73 and the NAMs. A thorough analysis of this data revealed several features of TE-driven genomic changes in maize that are highly relevant in the field of plant evolutionary genomics.

1- the authors clearly show that only 10% of the SVs could be accounted for by a transposition event (SV = TE), while the rest of the variants encompass severl TEs and suggest that they are the result of TE elimination which leads the authors to conclude that maize genome is decreasing in size through this TE elimination process.

2- Through an elegant method consisting in targetting low SNP regions, the authors identify young insertions and therefore putatively recently active TE families.

3- the analysis of SVs in genes show that non syntenic genes (ie genes that are not conserved across grass genomes) have a higher propensity to be deleted (or inserted).

I have no major objection with the publication of this paper as it stands given only minor modifications are made :

-l 102 change "transposons" by "transposable elements"

-l 234 -> what does "constrained proportion" mean ? please rephrase.

-l 218-224 -> I don't think one could draw any conclusion on the fact that some class II families exhibit more polymorphisms than others in the 26 NAMs. It just depends on whether a genome bears an active copy. The fact that monomorphic elements are still present in the genome strongly suggests that the corresponding family is still active, otherwise it would have been eliminated from the genome.

l 334-342 I was missing some information of the genetic relatedness between the NAMs. Low SNP regions suggest a common recent ancestry. But has this been quantified at some point ?

l 455-460 I think that the authors did not review this page of the manuscript before submission....."X Mb of putative..."

**Have all data underlying the figures and results presented in the manuscript been provided?**

Reviewer #1: **No: **See comments above

Reviewer #2: Yes

PLOS authors have the option to publish the peer review history of their article (what does this mean?). If published, this will include your full peer review and any attached files.

Reviewer #1: No

Reviewer #2: No

---

## [Decision Letter · Decision Letter 1]

14 Nov 2023

Dear Dr Munasinghe,

Thank you very much for submitting your Research Article entitled 'Combined analysis of transposable elements and structural variation in maize genomes reveals genome contraction outpaces expansion' to PLOS Genetics.  The revised version has clarified many issues raised, and provides much clearer insights. Issues raised on the supplementary data have been clarified as well, and the authors provide to the community a large set of data and scripts, including a shiny web app providing structural information for any region of the genome. These are important resources that will be of great use to the community, and the authors have made an important work on describing the available files in appropriate Readme files, which is highly valuable to reuse the data.

The manuscript was fully evaluated at the editorial level and by independent peer reviewers. The reviewers and myself appreciated the attention to an important topic but I identified some concerns that we ask you address in a revised manuscript.

In particular, after reviewing the data available, I pointed out that in the ‘Polymorphic TE calls’ files available on Dryad, there is no column classifying the TE copies as used in the manuscript: column 18 classifies TEs based on “shared, polymorphic, or ambiguous”, but there is no column with the ‘TE=SV, multiTE, etc.’ classification. Since the authors have this information, and since they are the data used to draw conclusions of the article, it would be nice that they share it with the community.

Besides this, I have a few minor points, mainly corresponding to typos:

p.2, l.24 : change ‘de-novo’ to ‘de novo’ in italics

p.3, l.2: The wording ‘pays a role’ is misleading. The authors find that the major process underlying structural variation is deletions. In this case, TEs are not actively involved in generating the diversity observed. Rather, they are passive elements on which the deletion process occurs. Please modify.

p.6, l.114-123. As pointed by reviewer in the original version, ‘transposon’ should be replaced by ‘transposable elements’ for more clarity.

p.11, l.217: vs should be in italics

p.15, l. 294: i.e. should be in italics

p.19, l.396 and l.397: this should be 139+79=218 and not 224. Please correct the typo.

p.22, l.452: via should be in italics

We therefore ask you to modify the manuscript according to the review recommendations. Your revisions should address the specific points made by each reviewer.

Again, we want to thank the authors for all the great work they provided.

Yours sincerely,

Clémentine VITTE, PhD

Guest Editor

PLOS Genetics

Justin Fay

Section Editor

PLOS Genetics

Reviewer's Responses to Questions

**Comments to the Authors:**

Reviewer #1: Authors provided a revised version of their manuscript which fulfills my requests. I do not have further comments to be addressed.

Reviewer #2: the authors have satisfactorily addressed the minor points I made in my frist review.

**Have all data underlying the figures and results presented in the manuscript been provided?**

Reviewer #1: Yes

Reviewer #2: Yes

PLOS authors have the option to publish the peer review history of their article (what does this mean?). If published, this will include your full peer review and any attached files.

Reviewer #1: No

Reviewer #2: No

---

## [Editor Report · Decision Letter 2]

28 Nov 2023

Dear Dr Munasinghe,

We are pleased to inform you that your manuscript entitled "Combined analysis of transposable elements and structural variation in maize genomes reveals genome contraction outpaces expansion" has been editorially accepted for publication in PLOS Genetics. Congratulations!

Yours sincerely,

Clémentine VITTE, PhD

Guest Editor

PLOS Genetics

Justin Fay

Section Editor

PLOS Genetics

Comments from the reviewers (if applicable):

Dear authors,

Thank you for submitting this new revision of the manuscript, and for adding files that will be useful to the community.

There is only one point on which we did not understand each others: on p.6, l.114-123, my point was to replace ‘transposon’ by ‘TE’ for more clarity. I do not want to hold the manuscript for this minor change, but I would appreciate if you could modify it on the proofs.

Thank you for all your work and for your valuable contribution to the generation of scientific knowledge!

Best regards,

Clémentine

**Data Deposition**

http://datadryad.org/submit?journalID=pgenetics&manu=PGENETICS-D-23-00407R2

**Press Queries**

---

## [Editor Report · Acceptance letter]

8 Dec 2023

PGENETICS-D-23-00407R2 

Combined analysis of transposable elements and structural variation in maize genomes reveals genome contraction outpaces expansion 

Dear Dr Munasinghe, 

We are pleased to inform you that your manuscript entitled "Combined analysis of transposable elements and structural variation in maize genomes reveals genome contraction outpaces expansion" has been formally accepted for publication in PLOS Genetics! Your manuscript is now with our production department and you will be notified of the publication date in due course.

With kind regards,

Zsofia Freund

PLOS Genetics

On behalf of:
